# Material Compatibility in 4D Printing: Identifying the Optimal Combination for Programmable Multi-Material Structures

**DOI:** 10.3390/polym16152138

**Published:** 2024-07-27

**Authors:** Matej Pivar, Urška Vrabič-Brodnjak, Mirjam Leskovšek, Diana Gregor-Svetec, Deja Muck

**Affiliations:** Faculty of Natural Sciences and Engineering, University of Ljubljana, Snežniška 5, 1000 Ljubljana, Slovenia; matej.pivar@ntf.uni-lj.si (M.P.); urska.vrabic@ntf.uni-lj.si (U.V.-B.); mirjam.leskovsek@ntf.uni-lj.si (M.L.); diana.gregor@ntf.uni-lj.si (D.G.-S.)

**Keywords:** 4D printing, material programming, multi-material structure, material characterization, thermal-responsive polymers

## Abstract

This study identifies the optimal combination of active and passive thermoplastic materials for producing multi-material programmable 3D structures. These structures can undergo shape changes with varying radii of curvature over time when exposed to hot water. The research focuses on examining the thermal, thermomechanical, and mechanical properties of active (PLA) and passive (PRO-PLA, ABS, and TPU) materials. It also includes the experimental determination of the radius of curvature of the programmed 3D structures. The pairing of active PLA with passive PRO-PLA was found to be the most effective for creating complex programmable 3D structures capable of two-sided transformation. This efficacy is attributed to the adequate apparent shear strength, significant differences in thermomechanical shrinkage between the two materials, identical printing parameters for both materials, and the lowest bending storage modulus of PRO-PLA among the passive materials within the activation temperature range. Multi-material 3D printing has also proven to be a suitable method for producing programmable 3D structures for practical applications such as phone stands, phone cases, door hangers, etc. It facilitates the programming of the active material and ensures the dimensional stability of the passive components of programmable 3D structures during thermal activation.

## 1. Introduction

Today, 4D printing is an emerging technology that builds on the principles of 3D printing, but with an added dimension of time. The fourth dimension refers to the ability of printed objects to change their shapes or properties over time in response to external stimuli such as temperature, humidity, or light. However, the choice of suitable materials for 3D printing is crucial for achieving the desired mechanical properties and functionality of the printed structures. Thermoplastic materials have gained significant attention due to their wide availability, ease of processing, and compatibility with various printing techniques. In recent years, there has been a growing demand for the development of complex programmed structures consisting of several different active elements with varying geometries, dimensions, and properties. These structures may involve the integration of electronic, mechanical, or biological components, which require precise and tailored printing processes to achieve the desired functionality [1,2,3,4,5]. However, the design and printing of such structures pose several challenges, such as the need for suitable materials with compatible properties, printing speed, the direction of filament loading, and layer thickness. To address these challenges, researchers have been investigating the most suitable and compatible combinations of thermoplastic materials for the 3D printing of complex programmed structures. This involves selecting materials with complementary properties, such as stiffness, strength, flexibility, and toughness, and optimizing the printing parameters to achieve the desired mechanical properties and functionality [6,7,8,9,10]. The development of such materials and printing processes has the potential to revolutionize the fabrication of complex structures for practical use in various fields, such as medicine, aerospace, and electronics. Therefore, understanding the principles of thermoplastic materials and their combinations for 3D printing is of great importance for advancing the field of additive manufacturing. Some of the most suitable and compatible combinations of thermoplastic materials for the 3D printing of complex programmed structures are [11,12,13,14,15]: ABS (acrylonitrile butadiene styrene) and PLA (polylactic acid)—ABS and PLA are two of the most widely used thermoplastic materials in 3D printing. ABS has good strength, toughness, and thermal stability, making it suitable for printing functional parts. PLA, on the other hand, is easy to print and has good dimensional accuracy. A combination of ABS and PLA can be used to print complex structures with varying mechanical properties;PET (polyethylene terephthalate) and TPU (thermoplastic polyurethane)—PET is a strong, stiff, and durable material, while TPU is a flexible and rubber-like material. A combination of these two materials can be used to print complex structures with varying mechanical properties, such as objects that require both rigidity and flexibility;PA (polyamide or nylon) and PVA (polyvinyl alcohol)—PA is a strong and durable material, while PVA is a water-soluble material that can be used as a support material during printing. A combination of these two materials can be used to print complex structures with overhangs or intricate geometries;PP (polypropylene) and PC (polycarbonate)—PP is a lightweight and flexible material, while PC is a strong and rigid material. A combination of these two materials can be used to print complex structures with varying mechanical properties, such as objects that require both flexibility and strength.

Thermoplastic materials for 4D printing depend on several factors, such as the desired response mechanism, processing requirements, and compatibility with 3D printing techniques [2,16]. A suitable combination of thermoplastic materials for printing more complex programmed 4D structures would require consideration of the following aspects: Stimulus-responsive properties—The materials should have responsive properties that can be triggered by specific external stimuli, such as temperature, humidity, or light. For example, shape memory polymers can exhibit a reversible change in shape or properties in response to temperature;Compatibility with 3D printing techniques—The materials should be compatible with the 3D printing techniques used for fabricating the structures. For example, materials with a low melting point can be printed using Fused Deposition Modelling (FDM) printers, whereas materials with a high viscosity require more advanced printing techniques like Stereolithography (SLA);Mechanical properties—The materials should have adequate mechanical properties to support the printed structures and withstand the desired stimuli. This includes factors such as stiffness, strength, and elasticity;Bonding properties: The materials should bond well with each other so as to avoid delamination or other bonding failures. This can be achieved by selecting materials that have similar melting temperatures or by using a chemical or physical bonding technique [17,18,19,20].

Several studies have explored the combination of different thermoplastic materials for 3D printing to improve the mechanical properties, print quality, and functionality of printed structures [12,21,22,23,24,25]. These studies have shown that blending different materials can improve the compatibility and performance of 3D printed structures, but the optimal combination depends on the specific application and requirements.

The materials obtained for 4D printing have a wide range of applications in various fields, each utilizing the unique ability of these materials to change their shape or properties over time in response to external stimuli. They can be used in fields such as medicine, aerospace, electronics, robotics, consumer products and even construction [26,27,28].

In medicine, 4D-printed structures made from shape memory materials such as PLA and HTPLA (High-Temperature Polylactic Acid) can be used to create drug delivery devices that release medication in response to body temperature or other stimuli. This can lead to more targeted and controlled drug delivery. Namely, biocompatible and biodegradable materials such as PLA and HTPLA can be used to print scaffolds that change their shape to mimic the dynamic environments of tissues to support tissue growth and regeneration [26]. Flexible and shape-changing materials such as TPU can be used to create customizable medical implants and devices that adapt to the body’s movements, improving comfort and functionality [26,27]. For space missions where minimizing the volume of the payload is critical, 4D-printed materials can be used to create components that expand or change shape once deployed in space to optimize storage and functionality. Materials such as HTPLA, which change shape when exposed to heat, can be used to develop aerospace components that adapt their properties to the thermal environment, improving their performance and durability [29,30]. The combination of materials such as PET and TPU enables the production of electronic components that are both rigid and flexible [31,32]. This is beneficial for the development of wearable electronics and foldable devices that can withstand mechanical stress while remaining functional. Further, 4D-printed materials with responsive properties can be used to develop sensors that respond to environmental changes such as humidity, light or temperature, and provide real-time data for various applications. The flexibility and shape memory of materials such as TPU and ABS can be used to develop robotic components that mimic natural muscle movements, enabling more versatile and adaptable robotic systems [31,32]. Shape-changing materials can be integrated into robotic systems to develop actuators and sensors that respond to environmental stimuli and improve the robots’ ability to interact with their environment. On the other hand, the combination of rigid and flexible materials, such as PLA and TPU, can be used to develop customizable garments that adapt to the user’s body shape and movements, providing a better fit and more comfort [33,34,35]. With 4D-printed materials, textiles can be produced that change their properties depending on temperature or humidity, creating adaptable clothing that improves comfort and functionality. In the construction industry, materials such as PP and PC can be used to print building components that adapt their properties to environmental conditions, such as windows that change their opacity depending on the incidence of light, or walls that adapt their insulating properties to temperature fluctuations. The use of shape memory materials can lead to the development of self-healing building materials that repair themselves when exposed to certain stimuli, increasing the longevity and durability of structures.

Our research propels advancements in additive manufacturing by developing multi-material printing techniques for programmable 3D structures, significantly enhancing design flexibility and functional adaptability. This is crucial for creating dynamic components suited for complex applications.

By meticulously adjusting 3D printing parameters, we have achieved control over the active materials, enabling the creation of structures with variable transformation radii. This capability allows these structures to dynamically respond to environmental stimuli, enhancing functionality in real-world applications.

We also utilized passive materials to bolster the structural integrity of these structures, ensuring dimensional stability during thermal activation and resistance to lateral deformations. This stability is vital for producing larger or more intricate designs, thus expanding the scope and scalability of 3D-printed designs.

A notable innovation in our research is the elimination of support structures during printing, facilitated by the flat geometry of the programmed structures. This significantly improves print quality, reduces material waste, and simplifies storage, handling, and transportation, addressing the logistical challenges of traditional 3D prints. Our work not only enhances current manufacturing practices but also lays the groundwork for future breakthroughs in 3D printing, offering new solutions to complex design and functional challenges across various industries.

## 2. Materials and Methods

### 2.1. Materials and Printing Procedures

Polylactide (PLA) was used as an actuator in the active part of the programmable 3D structures, which enables the shape change of the programmable 3D structures. Modified polylactide (PRO-PLA), acrylonitrile butadiene styrene (ABS) and thermoplastic polyurethane (TPU) were used as passive materials to serve as the constraint layer of the active and passive parts of the programmable 3D structures, which remain undeformed after thermal activation. All thermoplastic materials used in this study are among the most common filaments available for the 1.75 mm-diameter material extrusion process, with Plastika Trček (Ljubljana, Slovenia) serving as the manufacturer of all these materials.

The 3D test samples and programmable 3D structures were prepared for 3D printing using Slic3r software (version 1.3.1, open source) and produced using the ZMorph VX 3D printer (ZMorph S.A., Wroclaw, Poland). The print head was equipped with two brass extrusion nozzles with a diameter of 0.3 mm. To ensure the appropriate programmed state, specific printing parameters were assigned to the individual parts of the programmable 3D structures. For the active parts, an aligned rectilinear pattern with a fill density of 99% was used, which was aligned longitudinally in all layers. For the passive parts, a rectilinear pattern with an infill density of 100% was used. The layer height was constant at 0.2 mm for all materials and parts of the programmable 3D structures. The other constant 3D printing parameters assigned to each material are listed in Table 1. 

After 3D printing, the programmable 3D structures were thermally activated in temperature-controlled hot water at 80 °C for 15 min. The water temperature was set 20 °C above the glass transition temperature of the active PLA material [36]. After thermal activation, the programmed 3D structures were immediately cooled to room temperature.

### 2.2. Material Characterization

#### 2.2.1. Thermal and Thermomechanical Properties

The 3D-printed and temperature-activated samples were characterized using a Q200 DSC (TA Instruments, New Castle, DE, USA). For each sample, approx. 4–5 mg of material was analyzed under a nitrogen atmosphere in an aluminum pan over two heating cycles. The results of the first run indicate the thermal properties of the materials after 3D printing and after thermal activation, while the results of the second run show the thermal properties of the materials when the production history was eliminated. The measurements were performed over a temperature range from 0 °C to 250 °C for PLA, PRO-PLA, and ABS and from −70 °C to 250 °C for TPU polymer. The heating rate was 10 °C/min and the cooling rate 5 °C/min. The cold crystallization temperature (Tcc), melting temperature (Tm), glass transition temperature (Tg), cold crystallization enthalpy (ΔHcc) and melting enthalpy (ΔHm) were determined. The degree of crystallization (Xc) was calculated according to Equation (1).
Xc = (ΔHcc/ΔHo) × 100% (1)
where ΔHo is the enthalpy of 100% crystalline material [J/g] (∆Ho for PLA and PRO-PLA is 93.6 J/g and ∆Ho for TPU is 5.59 J/g, while ABS has no degree of crystallinity due to its amorphous structure). 

##### Dynamic Mechanical Properties

The dynamic mechanical properties of the 3D-printed samples were determined using dynamic mechanical analysis (DMA) with a DMA Q800 instrument (TA Instruments, New Castle, DE, USA). The 3D-printed samples had dimensions of 5 (x) × 60 (y) × 2 (z) and a programmable anisotropic structure. Measurements were performed in a double bending and tensile deformation mode at an oscillation frequency of 1 Hz, an oscillation amplitude of 10 μm and a heating rate of 3 °C/min over a temperature range from room temperature (23 °C) to a temperature at which all thermomechanical properties of the materials were stabilized [37]. The storage modulus (E′) and the tangent delta (tan δ), i.e., the damping factor, were determined in bending mode. The change in dimensions of the 3D-printed samples in the programmed direction during heating was determined in tensile mode. A constant low load of 0.01 N was used to reduce the thermal deformation of the samples due to external forces at high temperatures [38].

##### Dimensional Strain

The amount of stored pre-strain was determined by measuring the dimensional changes after the thermal activation of the 3D printed samples. A homogeneously laminated 3D-printed sample with the dimensions 4.5 (x) × 100 (y) × 3 (z) mm was used for the analysis. For each material, five 3D-printed samples were printed with an aligned rectilinear pattern oriented longitudinally in all layers to create an anisotropic structure, resulting in the anisotropic recovery of the filaments during thermal activation (Figure 1). The dimensions (width (x), length (y) and height (z)) of the 3D-printed samples were measured before and after thermal activation using a digital caliper (Mitutoyo, Japan). The measurements were performed with an accuracy of 0.01 mm. Unwanted lateral deformations after thermal activation were removed by flattening the programmed 3D structures so that the dimensions could be measured directly [39]. The dimensional strain (ε) was calculated using Equation (2).
ε = (L − L_0_)/L_0_ × 100% (2)
where L_0_ is the length of the 3D-printed sample before thermal activation and L is the length after thermal activation. 

#### 2.2.2. Apparent Shear Strength

When printing programmable 3D structures from multiple materials, it is important to achieve a sufficiently high bond strength between the materials to prevent delamination during thermal activation. During the thermal activation of programmed 3D structures, the different thermomechanical properties of the materials lead to an apparent shear strength between the materials. 

To determine the apparent shear strength between the materials, the single lap shear sample shown in Figure 2 was designed according to the guidelines of the EN 14869-2 and ASTM D3163-01 standards. Its geometry was adapted to material extrusion 3D printing technology to avoid printing support structures [40]. The analysis was performed on an Instron 5567 dynamometer (Instron, Norwood, MA, USA) at a test speed of 50 mm/min. For each material combination, five printed samples were tested under standard atmospheric conditions (22 °C, 50% RH). Prior to testing, all 3D-printed samples were conditioned in a standard atmosphere for 24 h. The comparative apparent shear strength for joints made from active PLA and passive materials (PRO-PLA, ABS, TPU) was determined for different printing sequences of active and passive material. The printing sequence of active and passive material is important for the production of the programmable 3D structures as it determines the direction in which the programmed 3D structures will transform. A test sample printed entirely with active PLA material was used as a reference to compare the apparent shear strengths of other material combinations.

#### 2.2.3. Experimental Determination of the Shape Transformation

The shape transformation of the heterogeneously laminated programmed 3D structures was determined by measuring the radius of curvature after thermal activation to determine the influences of different material combinations, their thermomechanical properties, the sequence of material printing and the material printing parameters on the shape transformation. The active part of the programmable 3D structures consisted of seven layers of active PLA material and one layer of passive material. The passive part consisted of eight layers of passive material. The length of the active part was 30 mm and the length of the passive parts was 10 mm. The dimensions of the programmed 3D structures and the constant layer height of 0.2 mm for all layers and materials were chosen on the basis of preliminary investigations. The structure of the programmable 3D structures is shown in Figure 3a.

After thermal activation, the heterogeneously laminated programmed 3D structures were cooled to room temperature to obtain the transformed shape. The images of the new shape of the programmed 3D structures were taken with a Nikon D750 (FX) camera (Nikon Europe, Amsterdam, Netherlands) perpendicular to the direction of the transformation. The radius of curvature of the transformation was determined by image analysis in Digimizer software (MedCalc Software Ltd., Ostend, Belgium) [41]. A smaller radius of curvature means a higher degree of transformation of the programmed 3D structures, while a larger radius of curvature means the opposite. The outer radius of the programmed 3D structures was measured as shown in Figure 3. For each material combination, ten samples were measured to determine the radius of curvature.

## 3. Results

### 3.1. Thermal Analysis

#### 3.1.1. The Crystallization Behavior and Melting Characteristics

The DSC heating thermographs for all materials are shown in Figure 4. Only one heating cycle, i.e., the second heating cycle, is shown in the thermographs, as no visible differences were observed between the first and second heating cycles. The DSC results are also summarized in Table 2.

Figure 4a,b show that PLA and PRO-PLA behave like semi-crystalline polymers. In both cases, the glass transition temperature is an indicator of the amorphous domain, while the melting temperature is a characteristic of the crystalline domain of the molecular structure.

The DSC curves of PLA showed three temperature transitions: glass transition, the cold crystallization peak (exothermic) and the melting peak (endothermic) at both heating cycles. The glass transition point was approx. 61 °C for the 3D-printed and thermally activated PLA (in the second heating sample). No differences in the temperature of the cold crystallization peak, the temperature of the melting peak or the degree of crystallization were observed for 3D-printed and thermally activated PLA (in the second heating cycle) (Table 2). In addition, no polymer degradation was observed up to 250 °C. The degree of crystallinity of the 3D-printed sample was 30.25 during the first heating cycle and decreased to 18.96% after thermal activation. In the second heating cycle, the degree of crystallinity for the 3D-printed sample and the thermally activated 3D-printed sample remained almost unchanged (approx. 27%).

The DSC curves for PRO-PLA are shown in Figure 4b. Again, the DSC curve shows a semi-crystalline molecular structure with three temperature transitions after both heating cycles—the glass transition, the cold crystallization and the double melting peak, except in the case of thermally activated PRO-PLA, where no exothermic cold crystallization occurred in the first heating cycle (also noted by Akhoundi et al. [42]). Compared to PLA, PRO-PLA has a slightly lower glass transition temperature (between 57 and 59 °C in the second heating cycle). The glass transition temperature of 3D-printed and thermally activated PRO-PLA also shows no significant differences (in both heating cycles). The peak of the cold crystallization temperature of PLA-PRO is also slightly lower compared to PLA: the modified PLA contains a filler that plays the role of a nucleating agent, which lowers the cold crystallization temperature. A similar phenomenon is also described by Yu et al. [43]. The double melting peak occurs in both 3D-printed and thermally activated 3D-printed PRO-PLA samples, and is a consequence of the crystallization behavior: during the formation of PLA, different crystals were formed that melt at different temperatures [44,45,46]. Compared to PLA, the first melting temperature of PRO-PLA decreased with the addition of filler, indicating that a larger number of less perfect crystals formed on the surface of the filler particles, which can be melted at lower temperatures [43]. The degree of crystallinity (in the second heating cycle) is 16.86% for the 3D-printed samples and 20.80% for the thermally activated samples, which is lower compared to PLA, probably due to the modified PLA structure. Again, no polymer degradation has been observed up to 250 °C.

The DSC curves of TPU, whose structure is composed of soft and hard segments, displayed some endothermic behavior by means of the disordering of crystallites. According to Frick and Rochman [47], the endothermic peaks at low temperature are related to the crystallites with a relatively short-range order, whereas the endothermic peaks at high temperature are caused by the disordering of hard segments of crystallites with a long-range order. At low temperatures, the glass transition between −35 and −39 °C was determined, reflecting the temperature relaxation of the soft segments ascribed to the co-operative and long-distance molecular motions. At temperatures between 30 and 100 °C, glass transition and enthalpy relaxation are related to the relaxation of the hard segments. The peaks at higher temperature could be related to the formation of hard segments of crystallites, influenced by both phase separation and phase mixing, as reported by Gallu et al. [48]. Double melting endotherming peaks of 3D-printed and thermally activated samples were determined: the first was found in the region 175.11–177.90 °C and the second one in the region 191.53–202.20 °C. These refer to the disordering of the two types of hard domain crystallites, as suggested by Frick and Rochman [47]. The crystallinity of TPU is very low, and estimated to be below 5% (Table 2).

The DSC curves for ABS are shown in Figure 4c. the ABS has an amorphous structure, and due to the absence of the crystalline phase, the DSC curve shows only one temperature transition (excluding the melting temperature) [49], i.e., the glass transition, which is practically indistinguishable in the case of the 3D-printed and the thermally activated sample, and is indicated at approx. 110 °C. 

#### 3.1.2. Dynamic Mechanical Properties

Figure 5 shows the curves of the bending storage modulus (E′) and damping factor (tangent delta) as a function of temperature for the 3D-printed samples, while their values at room temperature (23 °C) and at activation temperature (80 °C) are shown in Table 3.

As can be seen in Figure 5a, the curves of the bending storage modulus E’ of the 3D-printed samples for PLA and PRO-PLA had similar shapes. The storage modulus in the glassy region (i.e., below the glass transition) is the highest. In the glassy region, the material is brittle as its molecular chains are held in place without enough energy to overcome the molecular mobility barrier. In the rubbery state (above the glass transition temperature), the material becomes soft and flexible, and the storage modulus decreases. Due to the increased temperature, the molecular segments gain mobility and the molecular chains can slide past each other. A rapid drop in the storage modulus in a very narrow relaxation temperature interval for PLA and PRO-PLA indicates their predominantly amorphous structure. At room temperature, active PLA has the highest storage modulus E′ (3.123 GPa) due to its semi-crystalline structure, where the crystallites act as physical crosslinkers that have a direct influence on the elasticity of the materials (increased elasticity due to the cohesiveness of the structure). PLA therefore has the highest stiffness. Compared to PLA, PRO-PLA has a slightly lower elastic modulus (2.877 GPa) due to its lower crystallinity (and consequently lower crosslinking), which correlates with the degree of crystallinity (Xc) of the DSC analysis (Table 2).

At an activation temperature of 80 °C, at which PLA and PRO-PLA are in a viscous, rubbery state, PRO-PLA shows higher storage modulus values compared to PLA (0.070 GPa for PRO-PLA and 0.018 GPa for PLA; Table 3). But its storage modulus will have lost practically all of its size at room temperature. The connectivity of the structural intermolecular segments, as well as their crystallinity, are thus almost completely lost (by about 99% in the case of PLA and 97% in the case of PRO-PLA).

The bending storage modulus of TPU is given as 0.279 GPa at room temperature (lower compared to PLA and PRO-PLA) and 0.136 GPa at an activation temperature of 80 °C (higher compared to PLA and PRO-PLA). However, it drops by approx. 95% when the activation temperature is reached. 

The lowest degree of reduction in the storage modulus at the activation temperature is observed with ABS, namely, only 27%. This is understandable, since ABS remains in a glassy state in the thermal activation zone and has not yet exceeded the glass transition (Tg for ABS is around 110 °C, Table 2). The ABS molecular chains remain stiff (“frozen”) and do not yet have sufficient energy to ensure the structural mobility required for the flexibility of the material with a reduced storage modulus.

The damping factor (tangent delta) was used to investigate the damping behavior of the polymers. A higher damping factor indicates a higher mobility of the molecular chains, and is inversely related to the crystallinity and degree of cross-linking of the polymer. At an activation temperature of 80 °C, damping is low for all samples. The damping values at its peak are the highest for amorphous ABS without crystallinity. PRO-PLA has a lower damping factor than PLA at the peak values, which is due to the addition of fillers into the PLA matrix, as this inhibits the movement of the PRO-PLA chains due to the strong interactions between the filler and the polymer matrix. Similar results were also reported by Yu et al. [43].

The comparison of the expansion properties at a heating rate of 3 °C/min and a constant force of 0.01 N showed that the tested materials exhibited different thermal expansion behaviors, which is important for the fabrication of programmed 3D structures from multi-materials. The curve of changes in the initial length (Figure 6) of the active PLA sample shows a slight elongation of the sample by about 0.1 mm (0.3%) in the region of the glass transition temperature, followed by a drastic shrinkage from 70 °C and a renewed elongation with further heating in the rubbery region. The cause of the initial elongation of the sample in the glass transition region is probably the increase in free volume and the breaking of intermolecular bonds, which leads to a loosening of the PLA structure upon heating. When the resonance relaxations in the glass transition region cease, the intermolecular forces are restored, the free volume is reduced, and the material shrinks. After the release of the prestress, further heating contributes to the thermal expansion and the influence of the resulting viscous component, which consequently causes the tested sample to elongate. A similar thermal expansion behavior of the PLA sample in the programmed direction was also reported by other authors using the DIL 402 Expedis dilatometer [38].

The curve of changes in the initial length of passive ABS shows that the largest dimensional changes in the ABS sample are observed in its glass transition region, i.e., from 110 °C, while no significant shrinkage is observed in passive PRO-PLA over the entire range of the heating temperature. In the glass transition region of active PLA, both PLA and PRO-PLA are relatively dimensionally stable. As we have seen in the DSC analysis, PRO-PLA has shown very similar temperature transitions compared to active PLA, but higher thermal dimensional stability, which is due to the fillers in the PRO-PLA blend matrix. Therefore, when selecting thermoplastic materials for printing programmable 3D structures, not only must the thermal transitions of the materials be considered, but also the properties that influence the thermal dimensional stability. The curve of changes in the initial length of the passive TPU sample shows that the TPU sample lengthens slightly continuously in the range of the heating temperature between 0 and 120 °C. This is understandable, as the glassy relaxation transition is below the freezing point. As already mentioned, further heating above the glass transition point contributes to the thermal expansion of the sample, as it affects the formation of the viscous component, which consequently elongates the sample.

#### 3.1.3. Dimensional Strain

Table 4 and Figure 7 show a comparison of the dimensional strain of homogeneously laminated thermally activated 3D-printed samples made of different materials in all three directions. 

The thermally activated 3D-printed samples made of active PLA and passive PRO-PLA exhibit anisotropic thermomechanical behaviors. A maximum shrinkage of 13.09% in the longitudinal direction was calculated for the active PLA material, indicating that PLA has sufficiently programmable properties for 4D printing under the current 3D printing parameters. However, the slight expansion of the filaments in the transverse direction (ε2) and the pronounced expansion of the filaments in the vertical direction (ε3) should not be neglected, as this can also influence the dimensional stability of the programmed 3D structures. An anisotropic thermomechanical behavior was also observed with the passive PRO-PLA material. A shrinkage of 1.86% in the longitudinal direction was calculated, which is 11.23% less than the shrinkage of the active PLA material in the same direction. Due to the sufficiently large difference, PRO-PLA in combination with PLA is suitable for the production of the active parts of programmable 3D structures. In the case of ABS and TPU, an anisotropic structure was also produced in the 3D-printed samples without any major dimensional changes in the longitudinal and transverse directions. In terms of height, however, extremely small changes were observed, which can be attributed to measurement errors. DMA and DSC analyses showed that these two materials are more temperature-stable and do not exhibit anisotropic thermomechanical behaviors at an activation temperature of 80 °C.

### 3.2. Apparent Shear Strength

When printing programmable 3D structures from materials that have different linear thermal expansion coefficients, it is particularly important that a sufficiently high interfacial adhesion is achieved between the adherent materials so that delamination does not occur during the thermal activation process. It is also important that good interfacial adhesion between the two materials is achieved with a different print sequence of the active and passive materials. This allows the fabrication of more complex programmed 3D structures that can transform in both directions (bi-lateral).

A symmetrical geometry of the printed test samples (Figure 2) ensures that the contact area between the bonded surfaces is consistent, the load applied during the lap-shear test is evenly distributed across the bonded interface, and at the same time, uneven stress concentration is minimized. Very low standard deviation in the measured values confirms the accuracy of the results obtained (Figure 8a). Despite the symmetrical geometry of the samples, the order of printing significantly impacts the interlayer adhesion and resultant apparent shear strength. The differences in thermo-mechanical properties, cooling rates and surface characteristics between PLA, PRO-PLA, ABS and TPU determine the interlayer adhesion. 

Fractures of test samples obtained via a single-lap shear test are shown in Figure 8. A test sample made entirely from active PLA, which served as the reference, achieved an apparent shear strength of 26.82 MPa. Test samples made with both combinations of active PLA and passive PRO-PLA exhibited similar high interfacial adhesion, with similar strength values. In the case of printing PLA/PLA and PRO-PLA/PLA, fracturing of the material occurred, whereas in the case of printing PLA/PRO-PLA, a partial delamination also occurred, with traces of overprinted material on the surface of the overlaid part of the test sample. As both materials are made from the same polymer (PLA) and have very similar thermomechanical properties and thermal transitions, as seen from the DSC and DMA analyses, interactions between and the bonding of molecular chains in the interface region take place. When PLA is printed first and PRO-PLA later (PLA/PRO-PLA combination), the apparent shear strength is slightly lower (15%) due to partial delamination. Nevertheless, the interfacial adhesion is still high enough to prevent delamination during thermal activation.

The lower interfacial adhesion could be related to the surface characteristics. The surface of PLA is smoother and gives fewer options to interconnect with the overprinted PRO-PLA. When PRO-PLA is printed first, the surface of filaments is rougher, and because of the larger specific surface, the overprinted molecular chains of PLA, with higher mobility, can bond to a greater extent, resulting in better interfacial adhesion.

The apparent shear strengths of PLA/ABS, ABS/PLA and PLA/TPU are lower than the reference, by more than half of its value. No fracture of adherent materials or delamination is seen on test samples (Figure 8); failure occurred in the joint and not in the substrate. Such low strength of the adhesively bonded materials can lead to the delamination and separation of materials during thermal activation. 

For good interlayer adhesion, chemical affinity is an important factor. Unlike the combination of PLA and PRO-PLA, which have good chemical affinity, PLA and ABS have poor chemical affinity. Due to their different chemical structures and differing polarities, PLA and ABS do not adhere well at the molecular level. The lack of strong interfacial adhesion leads to weak bonding between layers.

However, differences in apparent shear strength also appeared depending on the sequence of printing of the materials. PLA was printed at a temperature of 195 °C and ABS at a temperature of 260 °C. In the PLA/ABS combination, a lower apparent shear strength was obtained than in the case of the ABS/PLA combination. In the first case, the adhesion strength was lower because ABS, which is known for its very large shrinkage during cooling, is placed on the cooler PLA material, resulting in a lower mobility of the molecular chains, and good interfacial adhesion cannot be achieved. The adhesion could be improved by increasing the temperature of the build plate and the printing temperature of the PLA, but this would affect the prestress state of the active PLA material and its ability to transform.

When PLA is printed first, the lower temperature and quicker solidification of PLA create a less favorable surface for bonding with ABS. PLA cools and solidifies relatively quickly, and consequently the surface does not provide sufficient wetting for the molten ABS or PRO-PLA. A more rigid layer does not allow as much interlayer diffusion, resulting in weaker bonding and higher residual stresses. When ABS is printed first, it provides a better bonding surface for PLA, resulting in higher apparent shear strength.

When printing TPU/PLA, the second printed material, PLA, did not adhere and bond with the TPU, even when reducing the printing speed to make the loading more accurate and reducing the cooling speed to allow the longer mobility of molecular chains. Complex programmed 3D structures that would transform in both directions cannot be printed for this combination because of adhesive failure.

### 3.3. Experimental Determination of the Shape Transformation

The experimental determination of the shape transformation of the heterogeneously laminated programmed 3D structures showed that the sequence of printing of active and passive material influences the direction of the shape transformation (Figure 9). In the case of the first printed active PLA and the second printed passive material, the programmed 3D structures are transformed in the convex direction (downwards) with respect to the plane of the build plate; in the reverse sequence of printing the materials, they are transformed in the concave direction (upwards). The direction of the transformation is determined by the thermomechanical properties of the materials. The DMA analysis of the expansion properties and the analysis of the thermal shrinkage of the materials has shown that the active PLA material undergoes significantly greater thermomechanical shrinkage in the range of the activation temperature than all passive materials. When the programmable 3D structures are activated, the active PLA material shrinks more than the passive material, which in this case serves as a constraint layer, meaning that the programmed 3D structures transform in the direction of the active PLA material.

Figure 9, showing the transformed programmed 3D structures also shows that different thermomechanical properties and different parameters of the passive materials have an influence on the radius of curvature (Table 5), as different material combinations achieve different radii of curvature with the same geometric structure of the programmed 3D structures and the same activation conditions. The largest transformation among all material combinations with a radius of 8.86 (±0.105) mm is achieved by combining the first printed active PLA and the second printed passive PRO-PLA material. When the printing sequence is reversed, the materials achieve a slightly smaller transformation with a radius of 9.62 (±0.141) mm. The differences in the size of the radius depend on the printing sequence of the active PLA material. In the first case, in which the active PLA material is deposited directly onto the build plate heated to 30 °C, the material cools faster and maintains a higher stress state than in the second case, in which the active PLA material is deposited onto a pre-printed layer of PRO-PLA, which is significantly warmer than the build plate. As a result, the active PLA cools down more slowly and the stress state in the deposited filaments is consequently lower, leading to less deformation. Other researchers have come to the same conclusions [50].

By adjusting the printing speed, we found that a higher pre-stress in the PLA material leads to more pronounced transformations upon thermal activation. Specifically, when active PLA is layered over the pre-printed PRO-PLA material, the structures exhibit a radius of curvature of approximately 9.37 (±0.226) mm, a slightly lower deformation compared to structures containing the passive TPU material, which results in a larger radius of 12.10 (±0.377) mm.

Dynamic Mechanical Analysis (DMA) revealed that the transformation potential of 3D structures is influenced by the storage modulus of passive materials at the activation temperature of the active material. ABS displayed the highest storage modulus (1.314 GPa), leading to minimal transformation due to its resistance to bending, a result also supported by previous research [51]. In contrast, PRO-PLA, with the lowest modulus (0.070 GPa), facilitated the most significant transformation.

Furthermore, the coefficient of thermal expansion (CTE) and thermomechanical shrinkage also play crucial roles in the final transformation of the printed structures. We have noted that materials with similar CTEs and shrinkage rates, as well as optimized printing parameters like speed and temperature, can significantly mitigate differential shrinkage and deformation [52,53,54].

Lastly, the build plate and extruder nozzle temperatures were found to affect the stress states in the active PLA material, thus influencing its transformation. For instance, increasing the build plate temperature from 30 to 60 °C for the ABS material reduced deformation by diminishing the thermomechanical shrinkage in the active PLA.

## 4. Discussion

The research provides a characterization of materials (PLA, PRO-PLA, ABS and TPU) used in the 4D printing of programmable multi-material structures. The characterization was carried out by analyzing the thermal, thermomechanical and mechanical properties, and experimentally determining the radius of curvature of the programmed 3D structures.

The DSC analysis showed that both active PLA and passive PRO-PLA exhibit similar temperature transitions in a similar temperature range, but the latter has greater thermal dimensional stability due to the incorporation of structural fillers into the PLA matrix: PRO-PLA remained dimensionally stable in the thermal activation region, while active PLA shrank drastically. Since PLA-PRO also has the lowest flexural modulus in the temperature activation range, which gives it the greatest flexibility among all passive materials, it will be the most suitable choice for the shape transformation of multi-material structures in combination with the dimensionally unstable active PLA. TPU and especially ABS, with a high bending modulus in the temperature activation range, are not the best choice for passive parts, as their increased stiffness, which is due to their greater structural connectivity at the micron level (intermacromolecular connectivity), does not allow such a pronounced shape transformation of the multi-material structure as in the case of PRO-PLA (despite the extremely low crystallinity of TPU and the amorphousness of the ABS structure resulting from the DSC analysis). 

When analyzing the dimensional changes of the 3D-printed samples after thermal activation, it was found that the active PLA material tends to show the highest thermal shrinkage in the range of the activation temperature, while all other passive materials show significantly less or almost no shrinkage in the longitudinal direction. Further research will focus on how different printing parameters of the active PLA material can influence the dimensional strain and thus the control of the radius of curvature of the programmed 3D structures. 

When printing programmable 3D structures from multiple materials that have different linear thermal expansion coefficients, it is particularly important to achieve a sufficiently high shear strength between the materials to avoid delamination and unplanned transformations during the thermal activation process. It is also important to achieve good bond strength between the two materials with a different printing sequence of active and passive material. In this way, more complex programmed 3D structures can be produced that can transform in both directions (downwards and upwards). The combination of PLA as the active material and PRO-PLA as the passive material proved to be the best. Further investigations will focus on how thermal activation affects the apparent shear strength between two different active and passive materials.

By experimentally determining the radius of curvature of the transformation of programmed 3D structures, it was shown that the printing sequence of active and passive materials influences the direction of the transformation. In the case of the first printed active PLA and the second printed passive material, the programmed 3D structures are transformed in the convex direction (downwards) relative to the plane of the build plate, and in the reverse sequence of printing the materials, they are transformed in the concave direction (upwards). The direction of the transformation is determined by the thermomechanical properties of the materials, which was also proven by the DMA analysis of the expansion properties and the analysis of the determination of the thermal shrinkage of the materials. It has been shown that the active PLA material undergoes significantly greater thermomechanical shrinkage in the range of the activation temperature than all passive materials. When planning complex programmable 3D structures made of two materials, it is therefore necessary to consider the sequence of printing. 

## 5. Conclusions

In summary, our study successfully identifies the most effective combination of thermoplastic materials for the fabrication of multi-material programmable 3D structures capable of two-sided transformation. The combination of active PLA with passive PRO-PLA was particularly effective, exhibiting excellent apparent shear strength and optimal thermal–mechanical behavior conducive to complex transformations. These findings not only contribute to the advancement of material science in the field of 4D printing, but also pave the way for a wider range of purpose-built objects applied for everyday use, such as phone stands, phone cases, door hangers, etc. Future work will explore additional material combinations and refine printing processes to enhance the functional capabilities of programmed structures.

## Figures and Tables

**Figure 1 polymers-16-02138-f001:**
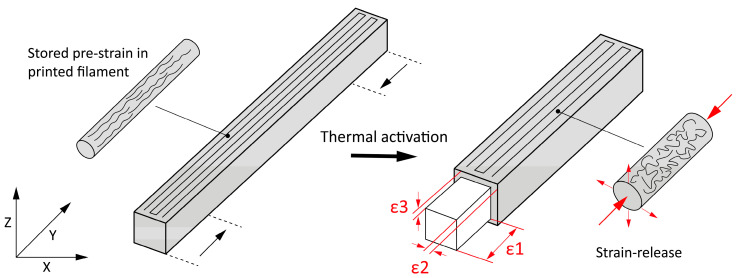
Schematic of a homogeneously laminated 3D-printed sample with a longitudinally aligned rectilinear pattern in all layers and anisotropic recovery of the programmable filament during thermal activation.

**Figure 2 polymers-16-02138-f002:**
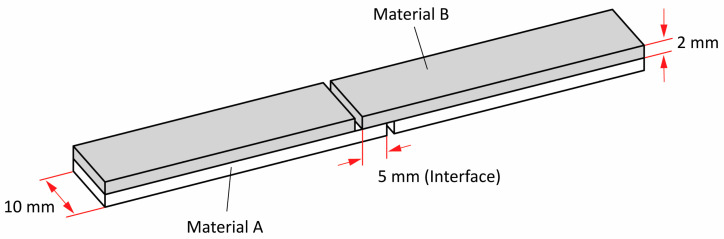
Schematic representation of the 3D-printed single-lap shear sample.

**Figure 3 polymers-16-02138-f003:**
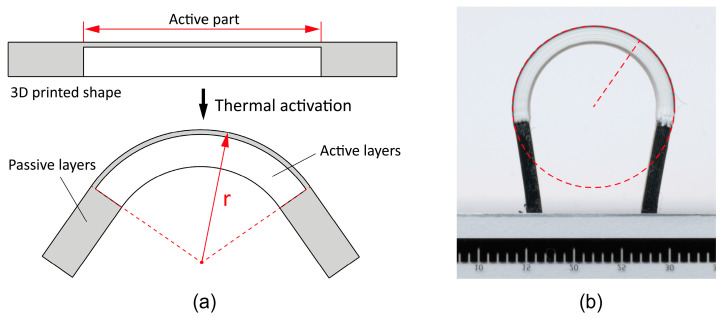
(**a**) Schematic representation of the 3D-printed programmable structure before and after thermal activation. (**b**) Determination of the outer radius of curvature of the programmed 3D structure.

**Figure 4 polymers-16-02138-f004:**
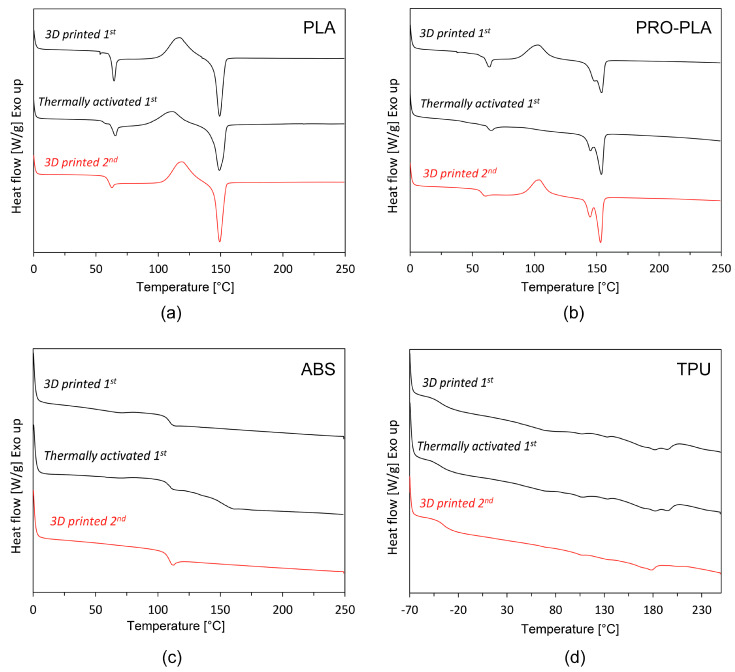
Comparison of DSC thermograms of first and second heating cycle before and after thermal activation: (**a**) PLA, (**b**) PRO-PLA, (**c**) ABS and (**d**) TPU.

**Figure 5 polymers-16-02138-f005:**
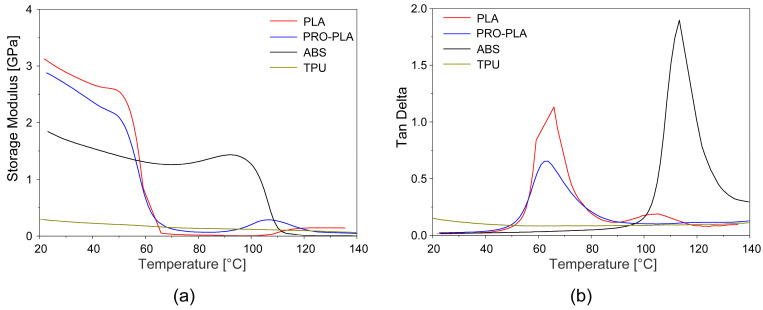
(**a**) Bending storage modulus (E′) and (**b**) damping factor (tangent delta) vs. temperature at 1 Hz of oscillation under DMA analysis.

**Figure 6 polymers-16-02138-f006:**
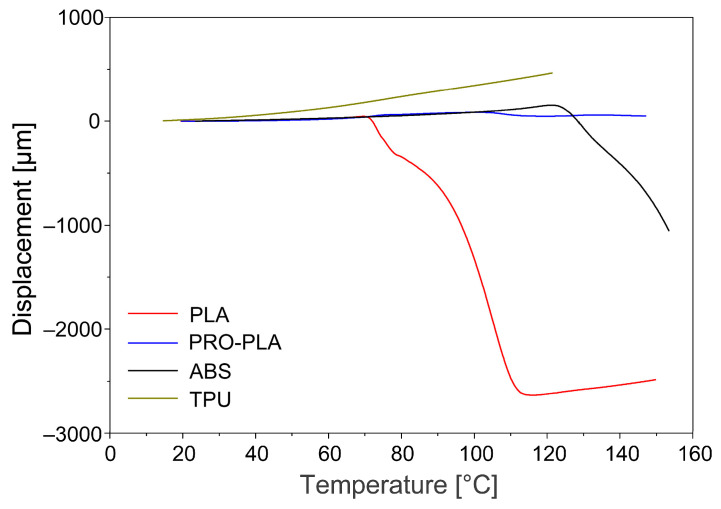
Change of the initial length of the 3D-printed samples vs. temperature.

**Figure 7 polymers-16-02138-f007:**
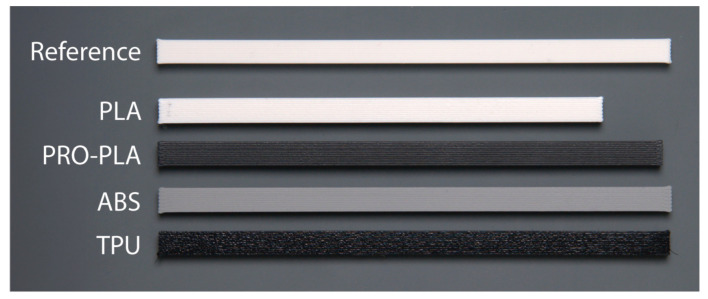
Effect of thermal activation on the longitudinal shrinkage of 3D-printed samples compared to a reference sample before thermal activation.

**Figure 8 polymers-16-02138-f008:**
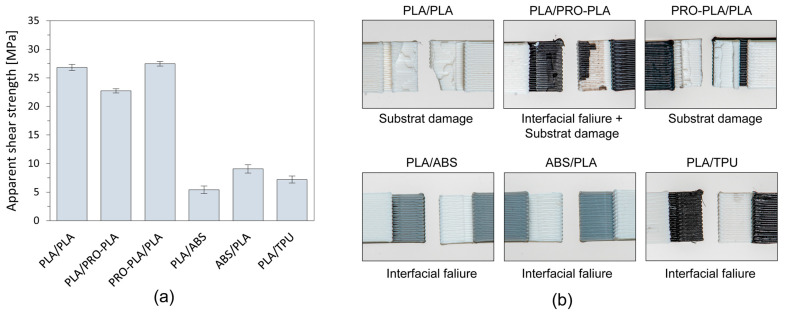
(**a**) Apparent shear strength of different combinations of materials. (**b**) Optical images of the damaged surface of different combinations of materials.

**Figure 9 polymers-16-02138-f009:**
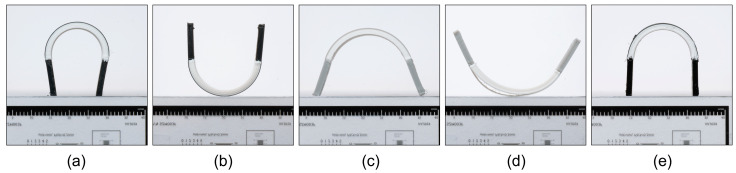
Heterogeneously laminated programmed 3D structures from different material combinations after thermal activation: (**a**) PLA/PRO-PLA, (**b**) PRO-PLA/PLA, (**c**) PLA/ABS, (**d**) ABS/PLA and (**e**) PLA/TPU.

**Table 1 polymers-16-02138-t001:** Printing parameters.

Material	Printing Temperature [°]	Build Plate Temperature [°]	Fan Speed [%]	Printing Speed [mm/s]
PLA	195	30	100	50
PRO-PLA	195	30	100	50
ABS	260	60	0	50
TPU	230	30	30	20

**Table 2 polymers-16-02138-t002:** Temperatures of glass transition (Tg), cold crystallization (Tcc), melting temperature (Tm), crystallization temperature (Tc), cold crystallization enthalpy (ΔHcc), melting enthalpy (ΔHm) and degree (rate) of crystallization (Xc) after first and second heating cycle.

Material	Sample	Heating Cycle	Tg [°C]	Tcc [°C]	Tm1 [°C]	Tm2 [°C]	ΔHm [J/g]	ΔHcc [J/g]	Xc [%]
PLA	3D printed	1st	63.11	117.09	149.36	–	24.55	28.31	30.25
		2nd	61.17	118.93	149.34	–	25.34	25.59	27.34
	Thermally activated	1st	64.09	111.07	149.08	–	26.48	17.75	18.96
		2nd	61.09	118.70	149.17	–	23.89	25.24	26.97
PRO-PLA	3D printed	1st	61.63	102.66	147.82	153.83	20.70	17.27	18.45
		2nd	58.83	103.61	144.52	153.04	23.07	15.78	16.86
	Thermally activated	1st	61.86	–	144.91	153.77	19.93	–	–
		2nd	57.37	105.89	143.50	151.71	22.23	19.47	20.80
ABS	3D printed	1st	109.17	–	–	–	–	–	–
		2nd	110.23	–	–	–	–	–	–
	Thermally activated	1st	110.21	–	–	–	–	–	–
		2nd	109.7	–	–	–	–	–	–
TPU	3D printed	1st	−38.01	–	177.30	191.53	13.86	6.487	3.29
		2nd	−35.47	–	175.11	201.61	9.247	7.582	3.85
	Thermally activated	1st	−37.90	–	177.90	192.21	16.41	6.891	3.50
		2nd	−36.17	–	175.80	202.20	9.782	5.130	2.61

**Table 3 polymers-16-02138-t003:** Bending storage modulus (E’) at room and at activation temperature and damping at activation temperature.

Material	E′ at 23 °C [GPa]	E′ at 80 °C [GPa]	Tan Delta (Damping Factor) at 80 °C
PLA	3.123	0.018	0.251
PRO-PLA	2.877	0.070	0.257
ABS	1.822	1.314	0.067
TPU	0.279	0.136	0.110

**Table 4 polymers-16-02138-t004:** Dimensional strain of 3D-printed samples after thermal activation.

Material	ε1 [%] (Longitudinal)	ε2 [%] (Transverse)	ε3 [%] (Height)
PLA	−13.09 (±0.115)	5.31 (±0.705)	12.14 (±0.263)
PRO-PLA	−1.86 (±0.064)	−0.24 (±0.519)	2.90 (±0.228)
ABS	0.01 (±0.004)	0.04 (±0.094)	0.20 (±0.183)
TPU	−0.01 (±0.010)	0.00 (±0.000)	−0.13 (±0.183)

**Table 5 polymers-16-02138-t005:** Radius of curvature of different material combinations.

	PLA/PRO-PLA	PRO-PLA/PLA	PLA/ABS	ABS/PLA	PLA/TPU
Mean [mm]	8.86	9.62	12.10	35.03	9.37
Std. Dev. [mm]	0.105	0.141	0.377	6.877	0.226
Max. [mm]	8.99	9.83	12.48	46.80	9.69
Min. [mm]	8.67	9.41	11.49	26.86	9.06

## Data Availability

The data are contained in this article.

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
