# Peer review of "Material Compatibility in 4D Printing: Identifying the Optimal Combination for Programmable Multi-Material Structures"

_polymers, 2024, doi:10.3390/polym16152138_

Round 1

Reviewer 1 Report

Comments and Suggestions for Authors

I don't know if active and passive thermoplastic materials are true thermotechnology. Please check.

The work lacks information about polymer manufacturers and their characteristics. It is absolutely unclear whether these are pure polymers or polymer compositions.

Information on the compatibility of various polymer fragments is mentioned very superficially. I think more discussion should be added on this topic.

The results are presented in subsection 3.3. should be summarized shortly.

Finally, it is necessary to discuss where the obtained materials can be used. I recommend considering a review https://doi.org/10.1002/tcr.202300217 where similar systems were disscussed.

Comments on the Quality of English Language

Minor editing of English language required

Author Response

I don't know if active and passive thermoplastic materials are true thermotechnology. Please check.

Thank you for your comment concerning the terminology of 'active' and 'passive' materials used in our study. These terms specifically delineate the roles materials assume in programmable 3D structures that are responsive to external stimuli such as temperature. 'Active' materials are characterized by their ability to undergo shape changes in response to such stimuli, while 'passive' materials provide essential structural support and maintain stability throughout the activation process. This distinction is integral to our research in 4D printing, which incorporates a time dimension into the design process.

To further clarify and support the terminology employed, we would like to reference several scientific publications that similarly distinguish between 'active' and 'passive' materials within the context of 4D printing and responsive material systems:

Theoretical stiffness limits of 4D printed self-folding metamaterials: link

Further examples where this terminology is applied can be found in the following articles:

  • Momeni, F., Mehdi Hassani N. S., Liu, X., & Ni, J. (2017). ', 122, pp. 42-79. This review offers an extensive examination of 4D printing technologies, discussing active and passive material systems and their responsive behaviors.
  • Ge, Q., Sakhaei, A. H., Lee, H., Dunn, C. K., Fang, N. X., & Dunn, M. L. (2016). 'Multimaterial 4D printing with tailorable shape memory polymers.' Scientific Reports, 6, 31110. This study demonstrates the utilization of shape memory polymers as active materials along with rigid, passive components to facilitate programmable actuation in 4D printed constructs.
  • Wang, Y.; Li, X. (2021). '4D-printed bi-material composite laminate for manufacturing reversible shape-change structures.' Composites Part B: Engineering, 219, 108918. Available at: link. The abstract describes the use of PLA for active layers and TPU for passive layers.

An additional reference where PLA and TPU are employed as active and passive layers, respectively, can be accessed here: link.

The work lacks information about polymer manufacturers and their characteristics. It is absolutely unclear whether these are pure polymers or polymer compositions.

Thank you. We have now included in the article the source and manufacturer for all thermoplastic materials used. The manufacturer is Plastika Trček, based in Ljubljana, Slovenia. This information has been added to the Chapter 2.1 "Materials and Printing Procedures".

In the article, the thermoplastic filament referred to as Pro-PLA is indeed the manufacuturer's trade name, HTPRO-PLA. We abbreviated the name for simplicity and ease of reference throughout the text. HTPRO-PLA is designed to be heat resistant up to 90°C and is highly suitable for mechanical parts due to its robustness. 

Information on the compatibility of various polymer fragments is mentioned very superficially. I think more discussion should be added on this topic.

Thank you. Information on the compatibility of various polymer fragments has been now expanded in Section 3.2, 'Apparent Shear Strength.'

The results are presented in subsection 3.3. should be summarized shortly.

Thank you. We have taken your comment into consideration and have extensively revised Subsection 3.3 of our manuscript. We have also added Table 5 to the manuscript, which presents the radius of curvature data along with mean values and standard deviations.

The subsection has been rewritten to include a concise summary of the results, ensuring that the findings are clearly presented and directly connected to the experimental setup and parameters discussed.

Finally, it is necessary to discuss where the obtained materials can be used. I recommend considering a review https://doi.org/10.1002/tcr.202300217 where similar systems were disscussed.

Thank you very much for the suggestion to supplement references. The usage of obtained materials has been added to the introduction. Additionally, all suggested references have been added.

Reviewer 2 Report

Comments and Suggestions for Authors

The title is not informative enough. It should be revised to better reflect the paper's objectives.

In line 98, HTPLA is not defined.

In line 102, the author mentions that structures made of PLA and HTPLA can respond to stimuli like humidity and light, which is not true. PLA and HTPLA are thermos-responsive shape memory polymers and only react to temperature changes.

In the materials section, the sources of the materials used are not mentioned.

In the introduction, the authors discuss HTPLA in detail, but for their experiment, they used Pro-PLA. Are these the same materials?

Some abbreviations are defined multiple times. For example, Tg is defined in lines 166, 179, and 277.

In Figure 8.a, the shear strength for samples PLA/ABS and ABS/PLA, as well as PLA/PRO-PLA and PRO-PLA/PLA, are different. Considering the symmetrical geometry of the printed samples for the shear test, could you provide more explanation for this result?

Author Response

The title is not informative enough. It should be revised to better reflect the paper's objectives.

Thank you for your recommendation regarding the title of our paper. We have considered your suggestion and agree that a revised title would more accurately reflect the focus and content of our research. Accordingly, we have updated the title to: »Material Compatibility in 4D Printing: Determining the Optimal Combination for a Programmable Multi-material Structure«.

In line 98, HTPLA is not defined.

Thank you. The name of the thermoplastic polymer material, HTPLA (High-Temperature Polylactic Acid), has now been included in text. It is provided next to the abbreviation in parentheses, consistent with the presentation of all other polymers mentioned in the document.

In line 102, the author mentions that structures made of PLA and HTPLA can respond to stimuli like humidity and light, which is not true. PLA and HTPLA are thermos-responsive shape memory polymers and only react to temperature changes.

Thank you for highlighting the inaccuracies in line 102 regarding the responsiveness of PLA and HTPLA to various stimuli. You are correct in pointing out that PLA and HTPLA are thermoresponsive shape-memory polymers and primarily respond to temperature changes, not to humidity or light as erroneously mentioned. We appreciate your attention to detail and have corrected this statement to accurately reflect the thermoresponsive nature of these materials.

In the materials section, the sources of the materials used are not mentioned.

Thank you. We have now included in the article the source and manufacturer for all thermoplastic materials used. The manufacturer is Plastika Trček, based in Ljubljana, Slovenia. This information has been added to the Chapter 2.1 "Materials and Printing Procedures".

In the introduction, the authors discuss HTPLA in detail, but for their experiment, they used Pro-PLA. Are these the same materials?

Thank you. In the article, the thermoplastic filament referred to as Pro-PLA is indeed the manufacuturer's trade name, HTPRO-PLA. We abbreviated the name for simplicity and ease of reference throughout the text. HTPRO-PLA is designed to be heat resistant up to 90°C and is highly suitable for mechanical parts due to its robustness.

Some abbreviations are defined multiple times. For example, Tg is defined in lines 166, 179, and 277.

Thank you. We have revised the text to ensure that abbreviations are defined only once, at their initial occurrence, and have removed redundant definitions from other parts of the text where they are not needed.

In Figure 8.a, the shear strength for samples PLA/ABS and ABS/PLA, as well as PLA/PRO-PLA and PRO-PLA/PLA, are different. Considering the symmetrical geometry of the printed samples for the shear test, could you provide more explanation for this result?

We appreciate your attention to the details in Figure 8.a. More explanation has been added to the text regarding the differing shear strengths observed for the samples PLA/ABS and ABS/PLA, as well as PLA/PRO-PLA and PRO-PLA/PLA.

Reviewer 3 Report

Comments and Suggestions for Authors

Manuscript Review

I believe that the research conducted is well-designed and organized. The topic is of current interest, being an active field of research today. In my opinion, the article deserves to be published in the journal Polymers; however, there are some points that should be improved to meet the quality required for publication.

General Comments:

  1. Clarity of Novelty and Application: At the end of the introduction, the novelty of the work and the application sought by creating different curvature radii are not clearly understood. It is advised to improve the paragraph from lines 131 to 142, clarifying these aspects.

  2. Conclusions: A paragraph of concise but clear conclusions is missing, highlighting the most important aspects obtained from the research.

  3. ABS in the Introduction: In the introduction, ABS is used as an example to manufacture programmed structures; however, in the manuscript, ABS is treated as a passive element. Clarify this fact.

  4. Discussion and Results: In my opinion, the content from lines 554 to 614 belongs more to the discussion than to the results.

Specific Changes or Improvements:

  1. Line 98: The acronym HTPLA is used but not defined. Include the definition.

  2. Materials and Methods Section:

    • It is necessary to include the commercial brands of the filaments used.
    • Explain the composition and properties of the so-called PRO-PLA, specifying how it has been modified and what fillers it contains, given that there are many modified PLAs.
  3. Equation 1: A 'c' is missing in the enthalpy of the "cold crystallization".

  4. Dimensional Strain: Include the nominal dimensions with which the specimens are designed.

  5. Point 2.2.3: Specify how many samples of each material combination were measured to determine the curvature radii.

  6. Figure 4: Include a label with the name of the material in each of the 4 graphs to facilitate reading.

  7. Redundancy: Eliminate lines 338 to 357 because they are repeated.

  8. Data Table: It is very important to introduce a table with the curvature radius data, along with their errors, instead of just listing them in the text.

Author Response

General Comments:

Clarity of Novelty and Application: At the end of the introduction, the novelty of the work and the application sought by creating different curvature radii are not clearly understood. It is advised to improve the paragraph from lines 131 to 142, clarifying these aspects.

Thank you. To address your request for clarification on the novelty of the work and the specific applications intended by creating different curvature radii, the whole paragraph was revised and rewritten.

Conclusions: A paragraph of concise but clear conclusions is missing, highlighting the most important aspects obtained from the research.

Thank you for your suggestion to include a concise concluding paragraph in our manuscript. Following your recommendation, we have added a clear and succinct conclusion at the end of the final chapter.

ABS in the Introduction: In the introduction, ABS is used as an example to manufacture programmed structures; however, in the manuscript, ABS is treated as a passive element. Clarify this fact.

Thank you. In the study, we use ABS as a passive material due to the appropriately chosen activation temperature of the programmed 3D structures. DSC analysis has shown that the ABS used in the study has a glass transition temperature of around 110 °C. DMA analysis of the initial dimension changes showed that it is precisely in the region of the glass transition that ABS begins to shrink. The activation temperature of 80 °C is too low to trigger stress release in the ABS material, which is why ABS remains undeformed and is used as a passive material. If the activation temperature were higher than 110 °C, ABS could be used as an active material that would trigger the transformation of the programmed 3D structures. 

Discussion and Results: In my opinion, the content from lines 554 to 614 belongs more to the discussion than to the results.

Thank you for your insightful comment regarding the placement and clarity of the content between lines 554 to 614. We have carefully reconsidered the structure of our manuscript based on your feedback. In response, we have rewritten the content to emphasize the empirical results more explicitly and to more directly connect the data to the specific configurations and settings used in our experiments.

Specific Changes or Improvements:

Line 98: The acronym HTPLA is used but not defined. Include the definition.

Thank you for bringing this to our attention. The name of the thermoplastic polymer material, HTPLA (High-Temperature Polylactic Acid), has now been included. It is provided next to the abbreviation in parentheses, consistent with the presentation of all other polymers mentioned in the document.

Materials and Methods Section:

It is necessary to include the commercial brands of the filaments used.

We have now included in the article the source and manufacturer for all thermoplastic materials used. The manufacturer is Plastika Trček, based in Ljubljana, Slovenia. This information has been added to the Chapter 2.1 "Materials and Printing Procedures".

Explain the composition and properties of the so-called PRO-PLA, specifying how it has been modified and what fillers it contains, given that there are many modified PLAs.

Thank you for your inquiry about the composition and properties of the material referred to as PRO-PLA in our article. For clarity and ease of reference, we abbreviated its name. PRO-PLA is indeed the manufacturer's trade name, HTPRO-PLA. HTPRO-PLA is specifically designed to be heat resistant up to 90°C and is highly suitable for mechanical parts due to its robustness. Its mechanical properties are comparable to those of ABS, but it offers greater toughness and reduced brittleness compared to standard PLA.

To further detail the modifications, an EDS (Energy Dispersive Spectroscopy) analysis was conducted on the PRO-PLA to determine the fillers it contains. The analysis indicated the probable presence of talc (magnesium silicate hydroxide) with the chemical formula Mg3Si4O10(OH)2, having a Si/Mg mass ratio of 1.54. Additionally, the comprehensive image area analysis revealed smaller proportions of aluminum (Al), calcium (Ca), and iron (Fe). This specific combination of fillers contributes to the enhanced mechanical and thermal properties of HTPRO-PLA.

Please note that the complete EDS analysis of the materials will be included in our next publication, and we prefer not to disclose full details at this moment to maintain the integrity of our forthcoming article.

Equation 1: A 'c' is missing in the enthalpy of the "cold crystallization".

Thank you for noticing that a 'c' was missing in the formula for the enthalpy of "cold crystallization." We have now corrected this error. The article has been updated to include the revised formula.

Dimensional Strain: Include the nominal dimensions with which the specimens are designed.

Thank you for your request for clarification regarding the nominal dimensions of the specimens used in our study. We have provided this information in Section 2.2.1 under the subheading "Dimensional Strain." In this section, the nominal dimensions with which the specimens are designed are clearly specified, ensuring transparency and replicability of our experimental procedures.

"A homogeneously laminated 3D printed sample with the dimensions 4.5 (x) x 100 (y) x 3 (z) mm was used for the analysis. "

Point 2.2.3: Specify how many samples of each material combination were measured to determine the curvature radii.

Based on your suggestion, we have updated Section 2.2.3 to include the specific number of samples measured for each material combination. The sentence was added at the end of the Section 2.2.3: "For each material combination, ten samples were measured to determine the radius of curvature."

Figure 4: Include a label with the name of the material in each of the 4 graphs to facilitate reading.

Thank you for your suggestion to label each graph in Figure 4 with the name of the material. We have added the material names to all graphs and have included these updated graphs in the manuscript. This change will facilitate easier reading and understanding of the data presented.

Redundancy: Eliminate lines 338 to 357 because they are repeated.

Thank you for pointing out the repeated text. We apologize for the oversight. We have now carefully reviewed the document and noticed that a substantial portion of the content was duplicated. We have removed the entire repeated section.

Data Table: It is very important to introduce a table with the curvature radius data, along with their errors, instead of just listing them in the text.

Thank you for your suggestion to include a table with the radius of curvature data. In response to your comment, we have added Table 5 to the manuscript, which presents the radius of curvature data along with mean values and standard deviations.

Round 2

Reviewer 1 Report

Comments and Suggestions for Authors

The authors have answered all my issues and paper can be accepted in its present form.

Comments on the Quality of English Language

Minor editing of English language required.

Reviewer 2 Report

Comments and Suggestions for Authors

Thanks for your efforts, the revised version is acceptable for publication.